# VISIONLOGIC: FROM NEURON ACTIVATIONS TO CAUSALLY GROUNDED CONCEPT RULES FOR VISION MODELS

## ABSTRACT

While concept-based explanations improve interpretability over local attributions, they often rely on correlational signals and lack causal validation. We introduce VISIONLOGIC, a novel neural–symbolic framework that produces faithful, hierarchical explanations as global logical rules over causally validated concepts. VISIONLOGIC first learns activation thresholds to convert neuron activations into a reusable predicate vocabulary and induces class-level logical rules from these predicates. It then grounds predicates to visual concepts via ablation-based causal tests with iterative region refinement, ensuring that discovered concepts correspond to features that are causal for predicate activation. Across different vision architectures such as CNNs and ViTs, it produces interpretable concepts and compact rules that largely preserve the original model's predictive performance. In our large-scale human evaluations, VISIONLOGIC's concept explanations significantly improve participants' understanding of model behavior over prior concept-based methods. VISIONLOGIC bridges neural representations and symbolic reasoning, providing more trustworthy explanations suited for safety-critical applications.

## 1 INTRODUCTION

Deep learning-based vision models have achieved remarkable success across numerous tasks, yet their *black-box* nature remains a major obstacle to trustworthy AI. This challenge has only intensified with the transition from Convolutional Neural Networks (CNNs) (LeCun et al., 1998; He et al., 2016) to Vision Transformers (ViTs) (Dosovitskiy et al., 2021), which introduce greater architectural complexity and opacity. To address this issue, a wide range of interpretability methods have been proposed. Among them, concept-based explanations (Nguyen et al., 2016; Bau et al., 2017; Kim et al., 2018; Ghorbani et al., 2019; Fel et al., 2023) have attracted particular interest because they uncover high-level semantic concepts rather than low-level attribution maps such as Grad-CAM and its extensions (Selvaraju et al., 2017; Wang et al., 2020; Jiang et al., 2021).

However, existing concept-based methods rely almost entirely on *correlational evidence* without *causal validation*, leading to potentially unfaithful or misleading explanations (Lopez-Paz et al., 2017; Zhang et al., 2023). For instance, TCAV (Kim et al., 2018) uses linear classifiers in activation space, while ACE (Ghorbani et al., 2019) applies clustering—both purely correlational approaches that lack rigorous causal foundations. As a result, these methods often conflate dataset biases with genuine model reasoning, such as associating the concept *pasture* with the class *cow*, a classic case where correlation fails to imply causation (Wu et al., 2023). Consequently, the concepts themselves may be spurious, leaving a fundamental methodological gap: the absence of *principled causal validation* for robust, interpretable concepts.

We address this gap with VISIONLOGIC, a novel neural-symbolic framework for generating faithful, interpretable explanations via global logical rules defined over causally validated concepts. Our approach operates in two stages. First, we transform high-level neuron activations into abstract predicates by learning activation thresholds and derive logical rules that approximate the model's class-level decision making. These predicates provide an intermediate symbolic representation that captures key aspects of the model's internal reasoning while remaining flexible and generalizable across input images. By converting neuron activations into predicates, we not only abstract the

model's computations but also create a structured foundation for reasoning about causally relevant concepts at a higher semantic level.

Second, we ground these predicates into high-level visual concepts using ablation-based causal tests. For each image, we start with an initial bounding box likely to influence the predicate, perturb the region with random noise or similar masking strategies, and check whether this flips the predicate's truth value. A transition from activation to deactivation provides causal evidence that the region is critical for the predicate. We then propose an efficient algorithm to iteratively refine the bounding box for more precise localization. For further refinement, segmentation methods such as Mask R-CNN (He et al., 2017) or SAM (Kirillov et al., 2023) are used to validate the intersection of the segmentation and refined box. Finally, the refined regions are consolidated across images within the same class to form consistent, causally validated visual concepts.

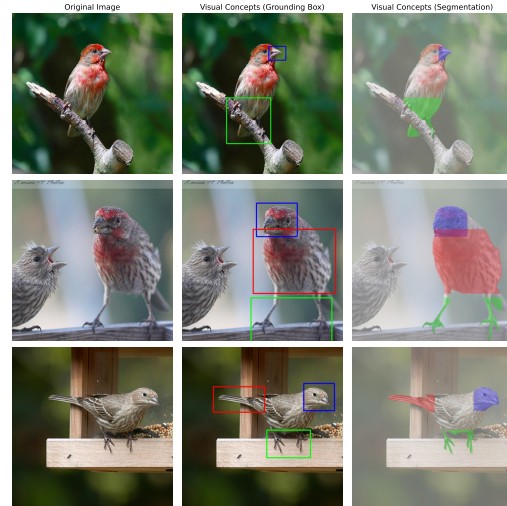

Figure 1: Causally validated concepts discovered by VISIONLOGIC, highlighted with bounding boxes and colored overlays.

The final result is a set of causally validated concepts and global logical rules that collectively provide transparent, faithful explanations of model behavior. For example, VISIONLOGIC discovers concepts such as *Beak* and *Claw* for the class *House Finch* in ImageNet (Deng et al., 2009), which are combined to form the logical rule: *Beak ∧ Claw ⇒ House Finch*, as illustrated in Figure 1. This rule reveals how the vision model leverages these causally validated visual concepts to make class-level predictions, providing interpretable insights into the model's decision-making process.

Our extensive human studies confirm that the concepts discovered by VISIONLOGIC significantly enhance understanding of the model's decision-making process compared to prior concept-based methods. Experiments on both CNNs and ViTs further demonstrate that VISIONLOGIC maintains strong predictive performance, achieving over 90% top-5 test accuracy on covered images, while providing explanations that are both causally grounded and human-understandable. *To the best of our knowledge,* VISIONLOGIC *is the first framework to deliver both causally validated concepts and interpretable logic-rule explanations.* We envision this as an important step toward bridging the gap between complex neural network representations and human-interpretable causal reasoning, offering trustworthy insights for high-stakes applications. Our contributions are summarized as follows:

- We propose VISIONLOGIC, a novel neural-symbolic framework that learns activation thresholds to form predicate-based abstractions and extracts logical rules over causally validated high-level visual concepts, bridging symbolic reasoning with neural representations.

- We develop an efficient, iterative refinement algorithm that precisely localizes causally relevant image regions using bounding-box adjustment and segmentation masks, ensuring accurate and consistent concept discovery.

- We conduct a large-scale human evaluation demonstrating significant improvements over state-of-the-art concept-based explanation methods in understanding the model's decision-making process through causally validated concepts and human-aligned interpretability.

- We empirically show that VISIONLOGIC largely retains the discriminative power of vision models with compact rules, and its grounded concepts are highly interpretable to humans, thereby providing a strong tool for understanding decision-making in vision models.

## 2    RELATED WORK

Existing approaches to interpreting vision models increasingly focus on *concept-based methods*, which aim to link internal representations to human-interpretable concepts rather than pixel-level

attributions (Selvaraju et al., 2017; Chattopadhay et al., 2018; Wang et al., 2020). Early studies explore the *hidden semantics* of neural networks by visualizing what individual neurons or layers encode. For instance, Nguyen et al. (2016) generate synthetic images that maximally activate specific neurons, while Mahendran & Vedaldi (2015) reconstruct inputs from intermediate feature maps to reveal the information preserved at different network depths. Network Dissection (Bau et al., 2017) further provides a systematic framework to quantify the alignment between hidden units and semantic concepts across diverse architectures. Although these methods offer valuable qualitative insights into model representations, they remain largely descriptive and lack systematic approaches to associate neurons with semantically meaningful concepts.

Subsequent work introduces *concept discovery methods* to establish explicit links between model activations and human-interpretable concepts. NET2VEC (Fong & Vedaldi, 2018) trains linear predictors on activation patterns to align feature maps with semantic labels, while TCAV (Kim et al., 2018) uses linear classifiers to score concept importance in activation space. However, all these approaches rely almost entirely on *correlational evidence* without any form of *causal validation*. As a result, they may capture spurious correlations between concepts and model decisions—for example, a concept might appear predictive of a class simply because both frequently co-occur in the training data, even if it plays no causal role in the model's reasoning (Wu et al., 2023).

More recent methods have attempted to refine concept discovery. For instance, ACE (Ghorbani et al., 2019) applies unsupervised clustering to extract concepts directly from activation patterns, while ICE (Zhang et al., 2021) improves upon ACE by introducing invertible concept-based explanations and leveraging Non-Negative Concept Activation Vectors to enhance interpretability and fidelity. CRAFT (Fel et al., 2023) further integrates sensitivity analysis into concept scoring to better measure how concept perturbations affect model predictions. Nevertheless, all these approaches still rely on unsupervised discovery techniques such as clustering or matrix factorization, which provide no causal guarantees (Lopez-Paz et al., 2017; Zhang et al., 2023). Consequently, the identified concepts themselves may be spurious, leaving a fundamental methodological gap: the lack of principled causal validation for robust and trustworthy interpretability in deep vision models.

## 3 THE VISIONLOGIC FRAMEWORK

VISIONLOGIC explains deep vision models by replacing the final decision layer with an interpretable program over *causally validated* concepts. The framework proceeds in three stages: (i) derive binary predicates from neuron activations, (ii) compose these predicates into class-wise rules and define an inference score, and (iii) ground predicates to image regions via *occlusion ablation*.

### 3.1 DERIVING PREDICATES FROM NEURON ACTIVATIONS

We begin with the final-layer activations $\mathbf{Z}(x) \in \mathbb{R}^d$ for an input $x$. For class $c$, the network computes the logit $\mathbf{F}^c(x) = \mathbf{W}^c \mathbf{Z}(x) + \mathbf{b}^c$, with $\mathbf{W}^c \in \mathbb{R}^d$ and $\mathbf{b}^c \in \mathbb{R}$. The term $\mathbf{W}_j^c \mathbf{z}_j(x)$ measures how much channel $j$ pushes toward class $c$ on this input. We sort channels by this per-example contribution so that rank 1 is the largest. Averaged over examples of class $c$, the *representativeness* of channel $j$ is captured by its expected rank; the most representative channel minimizes:

$$j^* = \arg\min_{j \in J} \; \mathbb{E}_{x \in X_c}\big[\mathcal{R}^c\big(\mathbf{W}_j^c \mathbf{Z}_j(x)\big)\big]. \tag{1}$$

Since $\mathbf{b}^c$ adds a constant to $\mathbf{F}^c(x)$, it cannot change the within-example ordering of contributions $\{\mathbf{W}_j^c \mathbf{Z}_j(x)\}_j$. The ranking is thus bias-invariant (Geng et al., 2022). For clarity, all classwise statistics in Section 3 use only training examples correctly classified by the base model, preventing predicate learning from contamination by misclassified instances.

We convert real-valued activations to *binary predicates* $p_j(x) \in \{0, 1\}$ that serve as logical atoms. Rather than fixing ad hoc thresholds, we learn per-channel thresholds $T_j$ and *sharpness* $s_j > 0$ (note: temperature $= 1/s_j$), which define a differentiable gate during training and a Boolean at test time:

$$\tilde{p}_j(x) = \sigma\big(s_j(\mathbf{z}_j(x) - T_j)\big), \qquad p_j(x) = \mathcal{I}\big(\mathbf{z}_j(x) \geq T_j\big). \tag{2}$$

The relaxed gate $\tilde{p}_j$ enables gradient-based learning of $(T_j, s_j)$; after training, we harden to $p_j$. We call a predicate *invalid* if its learned threshold makes $p_j(x) \equiv 0$ on all data; otherwise it is *valid*.

Note that some activation functions such as GELU (Hendrycks & Gimpel, 2016) can produce positive and negative responses with distinct semantics. To allow a single channel to encode two features, we define branch-specific predicates:

$$p_{j,+}(x) = \mathcal{I}(\mathbf{z}_j(x) \geq T_{j,+}), \qquad p_{j,-}(x) = \mathcal{I}(\mathbf{z}_j(x) \leq T_{j,-}), \qquad (3)$$

trained with the same relaxation as Eq. 2 using branch-specific $(T_{j,\pm}, s_{j,\pm})$.

A channel can be informative even when it is not the largest contributor on an example. We therefore require *both* high contribution rank and sufficient activation magnitude. We adopt a *single*, class-agnostic threshold $T_j$ shared across classes so that each predicate simply denotes "feature present" for neuron $j$; this deliberately *handles polysemanticity by reuse*—the same predicate may activate in multiple classes (Elhage et al., 2022). For class $c$ and input $x$, let $u_j^c(x) = \mathbf{W}_j^c \mathbf{z}_j(x)$ and $\mathcal{R}^c(u_j^c(x))$ be its within-example rank (1 is best). We define

$$p_{j,\leq k}(x) = \mathcal{I}\Big(\mathcal{R}^c(u_j^c(x)) \leq k \ \wedge \ \mathbf{z}_j(x) \geq T_j\Big), \qquad k \in \{1, 2, 3\}. \qquad (4)$$

During training, we replace the non-differentiable rank test with a *soft top-$k$* weight $w_j^c(x) \in [0, 1]$ computed with *SoftSort* (Prillo & Eisenschlos, 2020), which approximates the indicator $\mathcal{I}(\mathcal{R}^c(u_j^c(x)) \leq k)$. The relaxed gate is $\tilde{p}_{j,\leq k}(x) = w_j^c(x) \cdot \sigma(s_j(\mathbf{z}_j(x) - T_j))$. We instantiate a small set of rank windows $k \in \{1, 2, 3\}$ and impose *structured sparsity* (group lasso) across $\{p_{j,\leq 1}, p_{j,\leq 2}, p_{j,\leq 3}\}$ so the optimizer selects *at most one* $k$ per channel (i.e., the most predictive window), preventing predicate proliferation. The top-$k$ gate, combined with the shared threshold $T_j$, suppresses spurious activations and keeps the predicate vocabulary compact and easy to learn.

**Learning objective.** Given the predicate vector $P(x) = [p_1(x), \ldots, p_m(x)]^\top$, we train a lightweight linear head $f_{\text{rule}}(x) = \mathbf{W}_{\text{rule}}P(x) + \mathbf{b}_{\text{rule}}$ with the base network frozen. *At test time we do not use $f_{rule}$*; it serves solely to learn stable thresholds. The head supplies calibrated logits and, crucially, gradients that help place thresholds during learning:

$$\min_{\Theta_{\text{pred}}, \Theta_{\text{rule}}} \underbrace{\mathcal{L}_{\text{teach}}\big(f_{\text{rule}}(x), f_{\text{nn}}(x)\big)}_{\text{distill the frozen teacher}} + \underbrace{\lambda_T \|T - T^{(0)}\|_2^2 + \lambda_s \sum_j (s_j - 1)^2}_{\text{threshold/temperature stability}} + \underbrace{\lambda_{\text{use}}\Omega(\tilde{P})}_{\text{compact predicate set}}, \qquad (5)$$

where $\Theta_{\text{pred}} = \{T, s\}$, $\Theta_{\text{rule}} = \{\mathbf{W}_{\text{rule}}, \mathbf{b}_{\text{rule}}\}$, and $\tilde{P}$ collects the relaxed gates $\tilde{p}_j(x) = \sigma(s_j(z_j(x) - T_j))$. The distillation loss $\mathcal{L}_{\text{teach}}$ is the *Kullback–Leibler divergence* between the teacher and rule-head predictive distributions. The per-channel seed $T^{(0)}$ is initialized at a high percentile of $z_j(x)$ over influential training examples (we use the 0.8-quantile; influential = *SoftSort* top-$k$ by contribution with $k = 3$). We initialize $\mathbf{W}_{\text{rule}}$ with classwise normalized predicate frequencies (mean-centered by the global frequency per predicate) and set $\mathbf{b}_{\text{rule}}$ from class priors; values are then refined during learning. The stability terms keep thresholds near $T^{(0)}$ and temperatures near 1, preventing degenerate always-on/off gates. The compactness penalty $\Omega(\tilde{P})$ is a group-lasso over the rank variants $\{p_{j,\leq 1}, p_{j,\leq 2}, p_{j,\leq 3}\}$ for each channel, which selects a single rank regime and prevents predicate proliferation. More implementation details and hyperparameters are provided in the Appendix C.2.

*Observation.* Empirically, the learned thresholds $T_j$ often align with the $k{=}1$ specialization of the rank-aware predicate in Eq. 4: restrict to correctly classified examples from the class $c^\star(j)$ where neuron $j$ is most representative (i.e., has the lowest expected contribution rank), keep those instances $x \in X_{c^\star(j)}$ with $p_{j,\leq 1}^{(c^\star)}(x) = 1$ (i.e., $\mathcal{R}^{c^\star(j)}(u_j^{c^\star(j)}(x)) = 1$), and observe that $T_j$ tends to be close to the minimum $\mathbf{z}_j(x)$ over that subset. With *SoftSort* providing the soft top-$k$ proxy and structured sparsity selecting a single $k$ per channel, training recovers this heuristic in a data-driven way.

## 3.2 Learning logical rules and an inference score

Given the learned predicate vocabulary $P$, we induce symbolic rules and a *rank-based* inference score to explain the base model's predictions. For class $c$, we evaluate all predicates on each training example $x \in X_c$ that the base model classifies correctly, obtaining a binary vector $P(x) \in \{0, 1\}^m$. Each distinct vector defines a conjunctive clause that requires exactly the predicates that are true

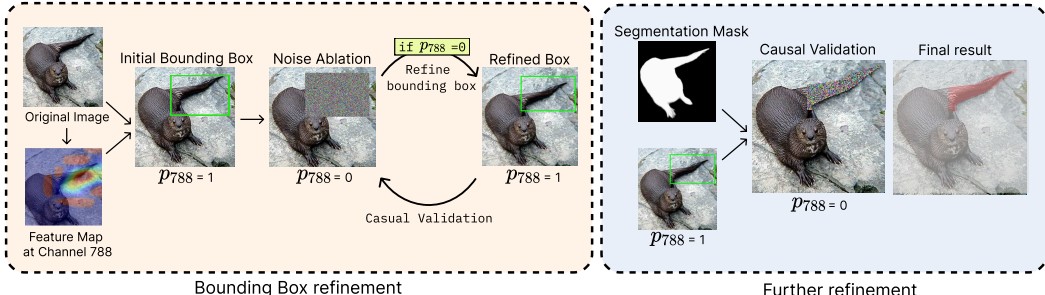

Figure 2: **Grounding predicates to visual concepts.** The orange panel illustrates how a bounding box is iteratively refined to capture the candidate region that causally influences an example predicate $p_{788}$ via noise ablation. The blue panel shows a further refinement step using segmentation masks.

and forbids those that are false. Joining all such clauses with disjunctions yields a DNF that exactly captures the training patterns of class $c$:

$$\forall x, \left( \bigvee_{\mathbf{v} \in \mathcal{V}_c} \left( \bigwedge_{i:v_i=1} p_i(x) \land \bigwedge_{i:v_i=0} \neg p_i(x) \right) \right) \implies \mathrm{Label}(x) = c, \qquad (6)$$

where $\mathcal{V}_c$ is the set of unique predicate patterns observed in $X_c$. Because exact matching is brittle on unseen data, we summarize class *characteristic strength* with a rank profile.

We build a class profile by counting predicate appearances across the class-$c$ clauses in Eq. 6 and sorting predicates by frequency to obtain a ranking $\mathcal{R}^c(p_i)$ (lower rank = more characteristic of $c$). For a test input $x$ with active predicates $P(x)$, we compute the explanation score:

$$S(x, c) = \frac{1}{|P(x)|} \sum_{p_j \in P(x)} \mathcal{R}^c(p_j), \qquad (7)$$

and predict with $\hat{c}(x) = \arg\min_c S(x, c)$. Intuitively, the chosen class is the one whose characteristic predicates best explain those active on $x$. Because predicates use a *single*, class-agnostic threshold $T_j$ per neuron, a predicate can fire in multiple classes (i.e., polysemanticity), but disambiguation comes from the class profiles $\{\mathcal{R}^c\}$: the same predicate typically has different ranks across classes, so the score $S(x, c)$ separates them.

*Observation.* Recall we initialize $\mathbf{W}_{\mathrm{rule}}$ with classwise normalized predicate frequencies (mean-centered by each predicate's global frequency). Since the class profile ranking $\mathcal{R}^c(\cdot)$ is induced by these frequencies, the resulting weights are, up to a positive monotone transform, equivalent to using inverse ranks (i.e., $\mathbf{W}_{\mathrm{rule}}^{c,i} \propto 1/\mathcal{R}^c(p_i)$). Consequently, maximizing $f_{\mathrm{rule}}^c(x)$ is monotone-equivalent to minimizing $S(x, c)$. A formal proof is given in Claim 1 (see Appendix C.2).

### 3.3 Grounding predicates to vision concepts

The final step grounds abstract predicates in the input space, linking logical atoms to interpretable visual features. Our binary predicate formulation enables principled causal reasoning—a key advantage over existing concept-based methods that rely on statistical correlations without establishing genuine causal relationships between visual features and model decisions.

Given an image $x$ and predicate $p_j$, we test whether a region is necessary for $p_j(x)$ by replacing that region with random noise[1] to obtain $x'$, then recomputing $p_j(x')$. A flip from activation to deactivation indicates that the region is causally important for $p_j(x)$. To find such regions efficiently, we initialize a bounding box intended to deactivate $p_j$ (typically large and image-covering). For CNNs, the initialization can be seeded from the feature map associated with $p_j$; for ViTs, it is aligned to the patch grid. We then iteratively refine the box until $p_j$ reactivates, following a procedure similar

---

[1]Other replacements, e.g., blurring, mean-filling, or blacking out, are also effective; in our experiments, blurring often performs best. We use random noise here for clarity and consistency in visualization.

Table 1: Utility scores in three application scenarios. The Utility benchmark measures how well explanations help users identify general rules that transfer to unseen instances. During training, participants are shown images along with the explanations and model predictions, and are asked to infer the underlying decision rules. At test time, the benchmark evaluates participants' accuracy in predicting the model's output on novel images. Higher Utility scores indicate that the explanations provide more useful information for understanding the model's behavior on new samples. For each scenario, the first and second best results are in **bold** and underlined respectively. *Our proposed method* VISIONLOGIC *achieves consistently higher Utility scores than prior approaches, with statistically significant improvements in the explanatory model's behavior across all three scenarios.*

| | Husky vs. Wolf | | | | Otter vs. Beaver | | | | Kit Fox vs. Red Fox | | | |
|---|---|---|---|---|---|---|---|---|---|---|---|---|
| Session n° | 1 | 2 | 3 | Utility | 1 | 2 | 3 | Utility | 1 | 2 | 3 | Utility |
| Baseline | 65.7 | 68.6 | 70.3 | 1.00 | 84.4 | 90.3 | 92.2 | 1.00 | 84.1 | 89.0 | 84.1 | **1.00** |
| Control | 55.3 | 63.6 | 70.0 | 0.92 | 85.1 | 88.3 | 92.9 | 1.00 | 80.8 | 79.2 | 79.2 | 0.93 |
| ACE | 60.4 | 71.1 | 74.6 | 1.01 | 80.4 | 85.7 | 90.5 | 0.96 | 80.6 | 83.2 | 76.2 | 0.93 |
| CRAFT | 55.5 | 60.8 | 65.3 | 0.89 | 86.3 | 90.9 | 90.9 | 1.00 | 76.8 | 81.8 | 76.8 | 0.92 |
| VISIONLOGIC | 74.8 | 90.0 | 91.0 | **1.25** | 96.8 | 98.4 | 99.2 | **1.10** | 84.1 | 84.5 | 82.9 | 0.98 |

to Geng et al. (2024); in some cases, only a few refinement steps are sufficient. The full algorithm is provided in Appendix A. Because the refinement is stochastic, multiple runs may yield different boxes; we retain the smallest successful box to improve precision. We also verify *sufficiency* by constructing an image with random noise everywhere except the candidate region and checking whether $p_j$ remains activated. This provides causal evidence that the candidate region influences $p_j$.

To better match object boundaries, we add a segmentation-based refinement step using off-the-shelf methods such as Mask R-CNN (He et al., 2017) or SAM (Kirillov et al., 2023). We intersect the segmentation mask with the refined box and repeat the intervention to confirm the expected predicate flip, thereby strengthening causal validity. Figure 2 illustrates the workflow. Finally, we aggregate validated regions across multiple images of the same class to form consistent, causally supported visual concepts, establishing a robust link between the concepts and $p_j$ and ensuring that the explanations faithfully reflect the model's decision-making.

## 4 EXPERIMENTS

### 4.1 HUMAN EVALUATION OF CAUSALLY GROUNDED CONCEPTS

**Setup.** We evaluate the practical utility of our proposed VISIONLOGIC, alongside prior concept-based methods ACE and CRAFT, in terms of *concept explanations* using the human-in-the-loop framework of Colin et al. (2022), which assesses how well explanations help participants understand a model's behavior across three real-world scenarios: (1) detecting bias in AI decisions (using Husky vs. Wolf classification), (2) identifying novel model strategies that are non-obvious to untrained observers (using Otter vs. Beaver classification[2]), and (3) understanding failure cases (using Kit Fox vs. Red Fox classification). Each scenario adopts the meta-predictor paradigm: during the training phase, participants study example images paired with explanation images and model outputs, then predict the model's output on unseen images without access to the corresponding explanations.

In total, 531 participants were recruited from Prolific (Prolific, 2024), in which 465 passed screening criteria aligned with Colin et al. (2022). We designed questionnaires for five conditions: (1) baseline (no explanation), (2) control (bottom-up saliency maps (Simonyan et al., 2014)), (3) ACE (Ghorbani et al., 2019), (4) CRAFT (Fel et al., 2023), and (5) VISIONLOGIC (ours), where ACE and CRAFT are considered state-of-the-art. To ensure fair comparison on the adapted datasets, all methods were re-run under identical experimental settings. Following prior work (Colin et al., 2022; Fel et al., 2023), we report the *utility score*, defined as participants' average test accuracy across the three sessions, normalized by the baseline; higher values indicate more effective human understanding of the model. Additional experimental details, including the participant recruitment process and the number of images used in each session and phase, are provided in Appendix F.1.

---

[2]This task replaces the legacy "Leaves" dataset from Colin et al. (2022) that is no longer available.

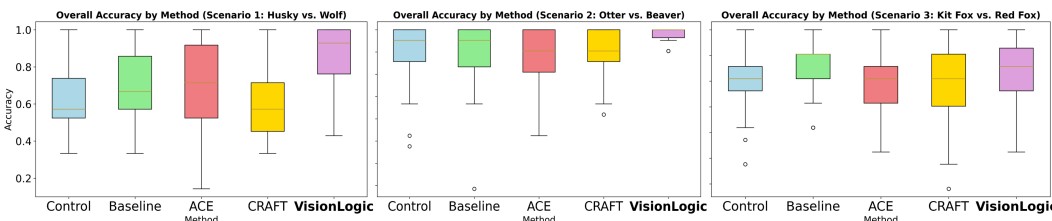

Figure 3: Data distribution for each explanation method in three scenarios. Our method VISIONLOGIC consistently enhances human understanding of model behavior over prior concept-based methods.

Table 2: Evaluation of VISIONLOGIC's rule-based explanations across vision models. The first column lists the models; the remaining columns report results for the proposed metrics. The #Valid metric gives the number of *valid* predicates; the following parentheses show the total predicate count.

| Model | #Valid | Complexity | Coverage (%) | Fidelity (%) | Top-1 Acc. (%) | Top-5 Acc.(%) |
|---|---|---|---|---|---|---|
| ResNet | 1944 (2048) | 9.49 | 83.48 | 75.42 | 69.27 | 93.53 |
| ConvNet | 1303 (2048) | 33.75 | 83.77 | 86.07 | 80.34 | 97.23 |
| ViT | 1465 (1536) | 42.63 | 80.48 | 87.55 | 80.70 | 97.38 |
| Swin | 1460 (1536) | 53.58 | 88.64 | 78.60 | 72.83 | 91.26 |

**Results.** Table 1 shows that VISIONLOGIC consistently outperforms ACE and CRAFT across all three scenarios. In the first two scenarios, VISIONLOGIC achieves utility scores significantly higher than 1, demonstrating clear benefits provided by our method to the participants when assisting them to infer model predictions. In the third scenario, we observe the same trend as prior work (Fel et al., 2023) where no existing method provides more effective information than Baseline. Nevertheless, our method shows substantial improvement over ACE and CRAFT; a utility score of 0.98 suggests that causally grounded concepts provide actionable guidance for understanding model failures.

To rigorously examine the performance gain of our method over Baseline, Control, ACE, and CRAFT in the evaluation shown in Table 1, we perform statistical tests on the collected data. We initially follow the procedure of Colin et al. (2022): an analysis of variance (ANOVA; (Scheffe, 1999)) followed by Tukey's honestly significant difference (HSD) test for pairwise comparisons (Tukey, 1949). However, we noticed that the assumption checks for normality and homogeneity, which are crucial to the validity of parametric test results (Öztuna et al., 2006), are absent from prior works (Fel et al., 2023). To this end, we performed a complete statistical testing procedure with assumption checks. Figure 3 shows that the data is highly skewed as test accuracy is upper bounded by 1.0, hence violating the normality assumption. Therefore, non-parametric statistics are more appropriate for testing whether there is any statistically significant difference between the five conditions listed.

A Kruskal-Wallis test (McKight & Najab, 2010) with a null hypothesis of *"All condition distributions are identica"* and an alternative hypothesis of *"At least one condition distribution differs"* at a significance level of 0.05 rejects the null hypothesis with $p = 3.4 \times 10^{-5}$, $8.83 \times 10^{-4}$ for the first two scenarios. We then utilize Dunn's test (Dinno, 2015) to analyze pairwise differences with Bonferroni correction (Weisstein, 2004) applied. In the first scenario, VISIONLOGIC is shown to be *significantly* better than all other conditions with $p = 3.03 \times 10^{-2}$, $4.00 \times 10^{-4}$, $2.41 \times 10^{-2}$, $p < 0.001$, respectively, suggesting that our method is effective in helping participants detect biases in the model. In the second scenario, VISIONLOGIC is shown to be *significantly* more effective than ACE and CRAFT with $p = 4.5 \times 10^{-3}$ and $3.09 \times 10^{-2}$ respectively, supporting its improvement over prior methods on the task of identifying unobvious visual clues. Full test details, including the statistical tests involved, assumption checks, corrections, and test statistics are reported in Appendix F.2.

## 4.2 EVALUATING LOGICAL RULE-BASED EXPLANATIONS

**Setup.** We evaluate the global logical rule–based explanations generated by VISIONLOGIC on deep vision models. To demonstrate generalizability across architectures, we cover four representative backbones: ResNet-50 (He et al., 2016), ConvNeXt-Base (Liu et al., 2022), ViT-B (Dosovitskiy et al., 2021), and Swin-T (Liu et al., 2021). VISIONLOGIC learns thresholds and logical rules on the full ImageNet-1k training set (Deng et al., 2009) (1,281,167 images) and is evaluated on the

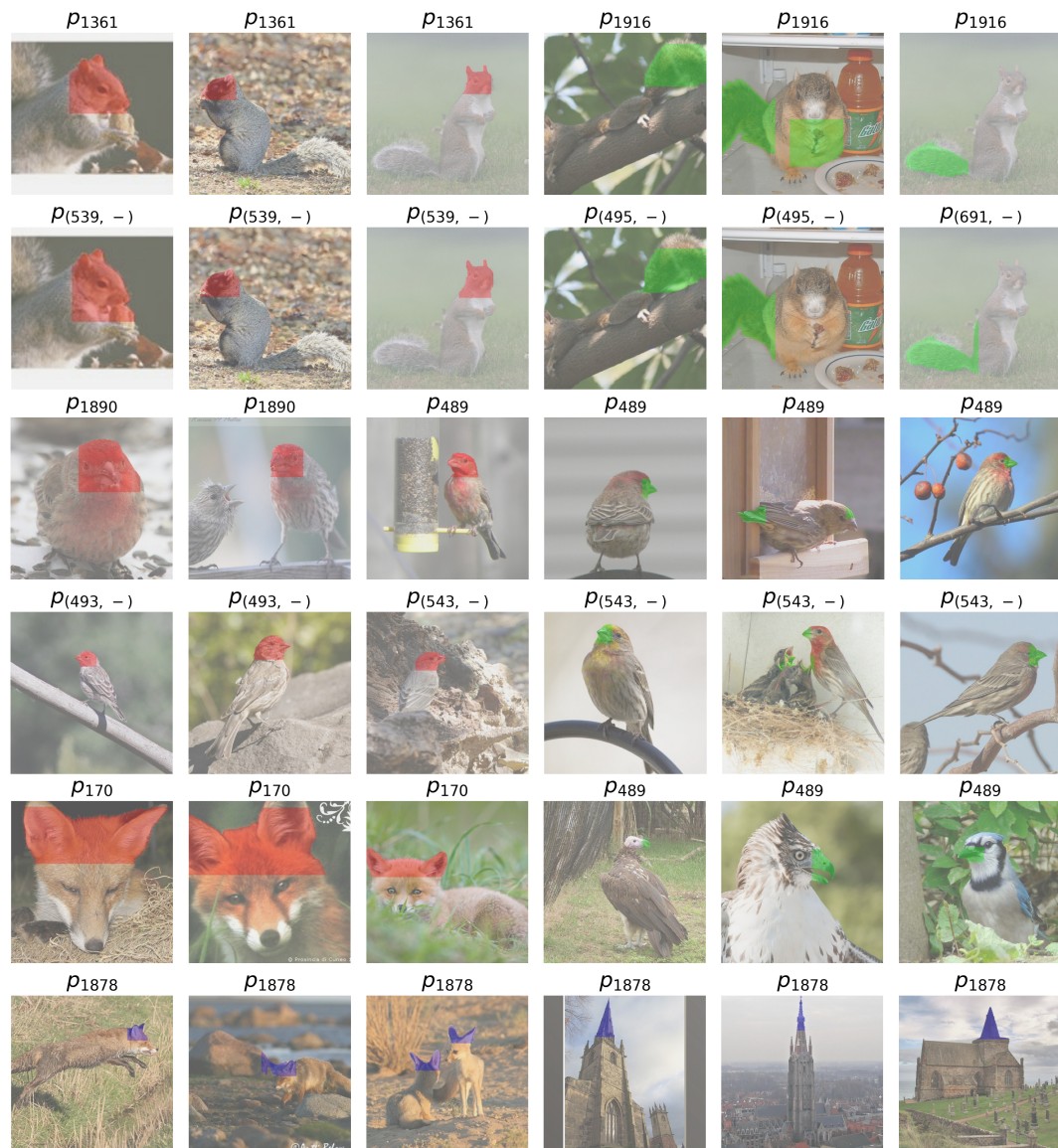

Figure 4: Hidden predicates grounded in visual concepts. In each image, the predicate appears above the frame, and the colored region in the image highlights the concept identified by VISIONLOGIC.

50,000-image validation set. To our knowledge, no prior method extracts *explicit, global* logical rules at this scale on modern backbones. Existing rule-extraction approaches (e.g., Cohen (1995); Zilke et al. (2016); Zarlenga et al. (2021); Hemker et al. (2023)) have only been applied to small datasets or shallow architectures, and have not scaled to state-of-the-art vision models. And the closest related effort (Jiang et al., 2024) relies on *indirect* evidence (e.g., counting sub-explanations) rather than explicitly extracting rules. *Thus, VISIONLOGIC provides, to the best of our knowledge, the first explicit, global interpretable rules for large vision models such as CNNs and ViTs.*

**Results.** Accordingly, we assess how well VISIONLOGIC preserves the base model's decisions and discriminative power, as well as the complexity of per-sample explanations, using metrics defined in Appendix D.1. Table 2 reports results across different base models under these metrics. Trained on the ImageNet training set, VISIONLOGIC exhibits strong generalization to the unseen validation set. VISIONLOGIC attains high *coverage* (80–89%) across all backbones while maintaining strong *fidelity* on covered images (76–88%). On covered, labeled images, VISIONLOGIC's rule-based predictions maintain competitive top-1/top-5 accuracy (ConvNeXt: 80.34%/97.23%; ViT: 80.70%/97.38%), indicating that the symbolic rules retain much of the base models' discriminative signal.

For a model-wise comparison, CNNs tend to produce shorter explanations than Transformers. For example, ResNet requires, on average, 9.49 predicates per image, whereas Swin requires 53.58, even though ResNet exposes more valid predicates overall (1,944 vs. 1,460). This aligns with recent findings (Jiang et al., 2024): CNNs exhibit more "disjunctive" (rule-like) behavior, while ConvNeXt and Transformers appear more "compositional," reflected in their longer clauses. Even 50 predicates, however, remain far more compact and interpretable than the thousands of hidden neurons driving base model decisions. Nevertheless, the rules may still appear complex to humans, and developing simplification methods that preserve predictive performance is an important direction for future work.

### 4.3 QUALITATIVE ANALYSIS OF VISUAL CONCEPTS

We present human-interpretable visual concepts encoded by predicates discovered in both ResNet and ViT. Each concept judgment is based on consistent visual inspection across many instances; Figure 4 shows representative cases. The 2nd and 4th rows are from ViT, the others from ResNet. Sampled concepts include *squirrel head*, *squirrel tail/paws*, *bird head*, *bird beak*, *fox ears*, and *church tops*.

**Polysemanticity.** We observe a many-to-many relationship between predicates and concepts (polysemanticity). A single predicate can deactivate when either of two distinct concepts is ablated, implying that one neuron-predicate may encode multiple concepts (within or across classes). Conversely, one concept can be encoded by multiple predicates, so masking its region can deactivate several predicates simultaneously. For example, in the 5th image of row 1, both the tail and paws of the squirrel are captured by $p_{1916}$ (ResNet). Predicate $p_{1878}$ simultaneously encodes *fox ears* and *church tops*; while semantically different, both share a triangular geometry, which may explain the reuse. Predicate $p_{489}$ (rows 3 and 5) frequently encodes *bird beak* across species (*jay*, *vulture*, *kite*, etc.), and recurs in local explanations for bird images, indicating an influential role in inference.

We also find cases where a single concept is captured by multiple predicates. In class *fox* (rows 5–6), both $p_{1878}$ and $p_{170}$ encode *fox ears*. A predicate may also attend to multiple instances of the same concept within an image: in row 6, image 3 (two foxes), $p_{1878}$ captures both pairs of ears. Such behaviors highlight the role of predicates as global concept detectors rather than merely local ones.

**Top-ranked predicates encode global structure.** Beyond the local and modular concepts discussed above, some predicates capture *global* object structure. Figure 10 (Appendix E) illustrates two classes (*church*, *squirrel*), where $p_{908}$, $p_{(498,-)}$, $p_{219}$, and $p_{(312,+)}$ (ResNet/ViT) encode the entire church or squirrel. Causally, masking any single part does not deactivate these predicates, but masking the whole object does. These global predicates are typically top-ranked within their classes and tend to be more class-specific, whereas local predicates are more frequently shared across classes. This pattern holds for both CNNs and Transformers and suggests a potential avenue for further rule simplification.

**CNNs vs. Transformers.** Similar concepts are found in both model families (see rows 1–2, where both capture the same concept on the same image). A key difference is that Transformers tend to involve *more* predicates per concept, whereas CNNs yield sparser, more distinct encodings. We hypothesize three factors: (1) CNN backbones expose a larger overall predicate set than ViTs; (2) activation functions (ReLU vs. GELU) induce different predicate sparsity via positive/negative branches; and (3) convolution versus attention imposes different inductive biases on concept formation. Validating these hypotheses and conducting in-depth analysis of learned predicates and rule structures, particularly differences between CNN- and ViT-based models, remains a direction for future work.

## 5 CONCLUSION

We introduce VISIONLOGIC, a novel neural-symbolic framework that produces faithful, hierarchical explanations as global logical rules over *causally validated* concepts, directly addressing the core limitation of prior concept-based methods that heavily rely on correlational statistics. VISIONLOGIC (i) learns activation thresholds to convert neuron activations into a reusable predicate vocabulary, (ii) induces compact, class-level logical rules over learned predicates that approximate the base model's predictions, and (iii) grounds predicates to visual concepts via ablation-based causal tests with iterative refinement. Across CNNs and ViTs, VISIONLOGIC largely retains models' discriminative power with compact rules. In human studies, it explains model behavior better than state-of-the-art concept-based methods, yielding clearer and more useful explanations. By unifying neural representations with symbolic reasoning, VISIONLOGIC offers trustworthy, actionable insight for high-stakes applications.

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

## A  THE BOUNDING BOX LOCALIZATION ALGORITHM

Algorithm 1 presents the detailed procedure for using bounding box localization to identify regions that significantly contribute to the computation of predicate $p_j(x)$. While the MASKREGION function in our implementation can replace the highlighted region with random noise[3], it also supports alternative masking strategies such as blurring, mean-fill, white-fill, and black-fill. We use blurring by default, as it performs most consistently in our experiments and aligns with our goal of non-destructive ablation. The GETFEATUREMAPREGION function thresholds the feature map at 15% of its maximum activation value, which results in connected segments of surviving pixels. It then draws a bounding box around the single largest segment (Zhou et al., 2016; Selvaraju et al., 2020). This yields a coarse initial estimate, which serves as a starting point for our iterative refinement algorithm, allowing it to converge toward more accurate and compact solutions.

---

**Algorithm 1:** BOUNDING BOX LOCALIZATION ALGORITHM

---

**Input:** Input image $x$, predicate $p_j$, model $M$, shrink factor $\lambda$, max attempts max_attempt
**Output:** Critical region $R$ for predicate $p_j$

1 **Function** InitialGuess($x$, $p_j$)
2     **if** $isCNN(M)$ **then**
3         $R \leftarrow$ GetFeatureMapRegion($p_j$)   /* Initialize using feature map if model is CNN */
4     **else**
5         $R \leftarrow$ LargeCentralBox($x$)     /* Use a large central box as default */
6     **return** $R$

7 **Function** RefineRegion($R$, $\lambda$)
    // Extract current region dimensions
8     $x, y, h, w \leftarrow R$
    // Generate a random sub-box with approximately $\lambda$ times the area of $R$
9     $R_{\text{new}} \leftarrow$ GenerateNewBox($R$, $\lambda$)
10    **return** $R_{\text{new}}$

11 **Function** LocateCriticalRegion($x$, $p_j$, $M$, $\lambda$, *max_attempt*)
12    $R \leftarrow$ InitialGuess($x, p_j$)       /* Start with an initial region guess */
13    **while** *True* **do**
14        $x' \leftarrow$ MaskRegion($x, R$)      /* Replace region $R$ in $x$ with noise */
15        $p_{\text{masked}} \leftarrow M(x')$
16        $p_{\text{crop}} \leftarrow M(\text{CropRegion}(x, R))$   /* Run model on cropped region alone */
17        **if** $p_{masked} < \tau$ **and** $p_{crop} \geq \tau$ **then**
18            refined $\leftarrow$ False
19            **for** $i \leftarrow 1$ **to** *max_attempt* **do**
20               $R_{\text{new}} \leftarrow$ RefineRegion($R$, $\lambda$)   /* Try a smaller random sub-region */
21               $x'' \leftarrow$ MaskRegion($x, R_{\text{new}}$)
22               $p_{\text{masked\_new}} \leftarrow M(x'')$
23               $p_{\text{crop\_new}} \leftarrow M(\text{CropRegion}(x, R_{\text{new}}))$
24               **if** $p_{masked\_new} < \tau$ **and** $p_{crop\_new} \geq \tau$ **then**
25                  $R \leftarrow R_{\text{new}}$
26                  refined $\leftarrow$ True
27                  **break**      /* Accept this refined region and continue */
28            **if** *not refined* **then**
29               **break**         /* Stop if no better sub-region found */
30        **else**
31            **break**       /* Initial region fails to meet constraints */
32    **return** $R$                 /* Return final critical region */

---

[3]Random-noise masks may introduce out-of-distribution artifacts and yield unstable model outputs.

## B    LLM USAGE

Large Language Models (LLMs) were used as a general-purpose assistive tool in the preparation of this work. Specifically, LLMs supported tasks such as refining the clarity of writing, suggesting alternative phrasings, and checking the consistency of technical terminology. They were **not** used for generating research ideas, conducting experiments, or producing original scientific contributions. All substantive research decisions, analysis, and results presented in this paper are the responsibility of the authors. The authors have carefully reviewed and verified all LLM-assisted text to ensure accuracy and originality.

## C    EXPERIMENTAL SETUP

All experiments are conducted on Ubuntu 22.04 LTS with an AMD EPYC$^{\text{TM}}$ 7532 (32 cores), 128 GB RAM, and a single NVIDIA A100 (40 GB). We use pretrained ImageNet-1k models to extract final-layer activations for all 1,281,167 training images. These activations are used to learn valid predicates and thresholds. We then compute class profiles and evaluate the induced logical rules on the ImageNet validation set (50,000 images) using the metrics in Appendix D.1. Hyperparameter choices and implementation details for the bounding-box algorithm (Algorithm 1) and predicate learning are discussed in the following subsections.

### C.1    BOUNDING BOXES

We localize predicate-supporting regions via iterative noise ablation using Algorithm 1.

**Initialization.** For CNNs, we first form an activation heatmap from the last-layer feature map. We binarize it at 15% of its maximum intensity, which results in connected pixel segments, and then draw the tightest axis-aligned bounding box around the single largest segment (Zhou et al., 2016; Selvaraju et al., 2020). If no pixels survive, we fall back to a centered box covering 90% of the image area. For ViTs, we initialize with a centered box covering 90% of the image, aligned to the patch grid.

**Refinement.** At each iteration, we propose up to 10 random shrinks of the current box using a shrink factor $\lambda = 0.9$ (uniformly sampling aspect ratio and position within the shrunken envelope). A proposal is *accepted* if ablating its region (replace with noise) flips the target predicate from active to inactive, $p_j(x) = 1 \rightarrow p_j(x') = 0$; otherwise it is rejected. We repeat until no accepted proposal exists. We run 5 independent trials per predicate (different random seeds) and keep the smallest accepted box across trials.

**Sufficiency check.** To verify that the retained region is sufficient to trigger the predicate, we paste the final box back into a noise canvas and confirm re-activation, $p_j(\hat{x}) = 1$.

**Further refinement.** We further refine the box using off-the-shelf segmentation models—SAM (Kirillov et al., 2023), Mask R-CNN (He et al., 2017), or ISNet (Jin et al., 2021). We intersect the predicted mask with the current box and then re-validate causality using the same ablation and sufficiency checks. Empirically, SAM and Mask R-CNN often produce fine-grained, part-level segments (e.g., an animal's ear or leg). While this does not affect the correctness of our pipeline, it can occasionally be cumbersome because it yields multiple disjoint regions to handle. By contrast, ISNet focuses on foreground–background separation and is therefore better aligned with our goal of isolating the entire foreground object (e.g., the whole car or animal) before computing the mask–box intersection. We hence mostly use ISNet with the default setting and hyperparameters in our experiments.

### C.2    THRESHOLD LEARNING PROBLEM

**Selecting influential examples.** For each channel $j$, we score contributions $u_j^c(x) = \mathbf{W}_j^c \mathbf{z}_j(x)$ per class $c$ and per example $x$. Influential examples for $j$ are those where $j$ is among the *SoftSort* top-$k$ contributors (Eq. 4) with $k = 3$. Concretely, we apply *SoftSort* to the vector $\{u_\ell^c(x)\}_{\ell=1}^d$ to obtain a differentiable top-$k$ mask $w_\ell^c(x) \in [0, 1]$, and declare $x$ influential for $j$ if $j$ belongs to the top-$k$ set under this mask (i.e., $w_j^c(x)$ is one of the $k$ largest values). *Intuitively, this selects examples where channel $j$ provides strong, class-relevant evidence, yielding stable, noise-resistant threshold seeds.*

**Seed threshold and sharpness.** The seed threshold and sharpness are

$$T_j^{(0)} \;=\; \text{Quantile}_{0.8}\Big\{\, z_j(x) \;\Big|\; x \text{ is influential for } j \,\Big\}, \qquad s_j^{(0)} = 1.$$

We initialize $T_j^{(0)}$ at a high percentile (0.8) of $z_j(x)$ over influential examples to anchor "feature present" in the activation tail, gain robustness to outliers, and start from a conservative, compact predicate set. We set $s_j^{(0)} = 1$ (logistic slope) and constrain $s_j \in [0.5, 5]$ during training for numerical stability.

**Objective and optimization.** We optimize Eq. 5 with Adam; learning rates are $10^{-3}$ for $\{T, s\}$ and $5 \times 10^{-4}$ for $\{\mathbf{W}_{\text{rule}}, \mathbf{b}_{\text{rule}}\}$, using batch size 512 and early stopping on validation KL (patience 5). Regularization uses

$$\lambda_T = 1.0, \qquad \lambda_s = 0.1 \;\; (\text{keeps } s_j \approx 1), \qquad \lambda_{\text{use}} = 5 \times 10^{-3},$$

with a group lasso over $\{p_{j,\leq 1}, p_{j,\leq 2}, p_{j,\leq 3}\}$ per channel to select a single rank window. We clip $T_j$ to the empirical range of $z_j$ (per channel, class-agnostic) and add a small $\epsilon = 10^{-6}$ inside the sigmoid to avoid saturation in mixed precision.

**Schedule and convergence.** We train for up to 30 epochs; $\mathbf{W}_{\text{rule}}$ and $\mathbf{b}_{\text{rule}}$ are warm-started from classwise normalized predicate frequencies (mean-centered by each predicate's global frequency), then jointly refined with $\{T, s\}$. Convergence is declared when validation KL improves by $< 10^{-4}$ or when the patience budget is exhausted.

**Hardening and test-time prediction.** After training, we harden $\tilde{p}_j(x) = \sigma(s_j(z_j(x) - T_j))$ to $p_j(x) = \mathcal{I}(z_j(x) \geq T_j)$ and discard $f_{\text{rule}}$. Test-time prediction uses the symbolic rank-based score:

$$\hat{c}(x) = \arg\min_c S(x, c),$$

i.e., the class whose characteristic predicates best explain those active on $x$ (Section 3.2).

**Notes and ablations.** Empirically, the learned thresholds $T_j$ often align with the $k{=}1$ specialization of the rank-aware predicate (cf. Observation), even though we train with candidate windows $k \in \{1, 2, 3\}$ and let structured sparsity select one per channel. In practice, $k{=}1$ is chosen for the majority of channels, with $k{=}2$ or $k{=}3$ retained for a minority of polysemantic channels that contribute reliably without being the single top contributor. Performance and rule sparsity are stable for $k \in \{2, 3, 4\}$; using $k{=}3$ as the candidate ceiling slightly improves recall of informative channels while keeping the predicate vocabulary compact. Using per-class thresholds harms transfer and readability; the class-agnostic $T_j$ yields stable, reusable predicates and defers disambiguation to the class profiles $\{\mathcal{R}^c\}$ via $S(x, c)$.

**Claim 1** (Monotone equivalence of rule head and rank score). *Initialize $\mathbf{W}_{rule}$ from classwise normalized predicate frequencies (mean-centered by each predicate's global frequency), which induces the same ordering over predicates as the class profile ranks $\mathcal{R}^c(\cdot)$. If, during training, the (training-only) head weights are set to a strictly decreasing affine function of rank, e.g.,*

$$\mathbf{W}_{rule}^{c,i} \;=\; \alpha_c \;-\; \beta\, \mathcal{R}^c(p_i), \qquad \beta > 0, \tag{8}$$

*and a class-independent bias $\mathbf{b}_{rule}$ is used (or scores are compared after subtracting class-specific constants), then*

$$\arg\max_c f_{rule}^c(x) \;=\; \arg\min_c S(x, c).$$

*Proof.* Let $A(x) = \{i : p_i(x) = 1\}$ be the active predicates and $m = |A(x)|$. Using Eq. 8 with a class-independent bias,

$$f_{\text{rule}}^c(x) = \sum_{i \in A(x)} \big(\alpha_c - \beta\, \mathcal{R}^c(p_i)\big) + \mathbf{b}_{\text{rule}} = m\, \alpha_c \;-\; \beta \sum_{i \in A(x)} \mathcal{R}^c(p_i) \;+\; \mathbf{b}_{\text{rule}}.$$

Since $m$, $\alpha_c$, and $\mathbf{b}_{\text{rule}}$ do not affect the ordering induced by the sum of ranks, maximizing $f_{\text{rule}}^c(x)$ over $c$ is equivalent to minimizing $\sum_{i \in A(x)} \mathcal{R}^c(p_i)$. By definition,

$$S(x, c) \;=\; \frac{1}{m} \sum_{i \in A(x)} \mathcal{R}^c(p_i),$$

so $\arg\max_c f_{\text{rule}}^c(x) \;=\; \arg\min_c S(x, c)$. $\qquad\square$

## C.3 Additional results on threshold learning

Figure 5–8 show the empirical distributions of learned per-channel thresholds $\{T_j\}$ across four architectures.

**(i) Transformers and ConvNeXt are bimodal and near-symmetric.** For ViT and Swin (Figure 7, 8), and for ConvNeXt (Figure 6), thresholds concentrate in two tight modes near $\pm 1$. This is consistent with (a) sign-aware predicates ($p_{j,+}$ and $p_{j,-}$) and (b) LayerNorm/GELU producing roughly zero-mean, unit-scale channel responses, so "feature present" naturally anchors away from 0 on both branches.

**(ii) ResNet is strictly positive and right-skewed.** For ResNet (Figure 5), thresholds are nonnegative and exhibit a heavy right tail. This aligns with ReLU activations (no negative branch) and localized high-energy features that occasionally require larger cutoffs; large $T_j$ outliers are infrequent but present.

**(iii) Conservativeness and the $k=1$ heuristic.** Across all models, thresholds sit well away from 0, indicating a conservative notion of "feature present." Qualitatively, many $T_j$ align with our $k=1$ heuristic (Section 3.1): for a given channel $j$, $T_j$ tends to be close to the minimum activation among correctly classified examples where $j$ is top-1 by contribution for its most representative class. This matches the view that training recovers a data-driven, architecture-stable cutoff.

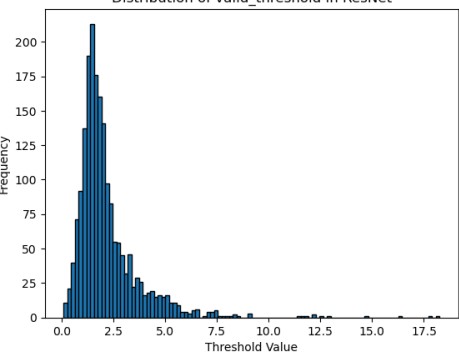

Figure 5: Distribution of learned thresholds for valid predicates in ResNet.

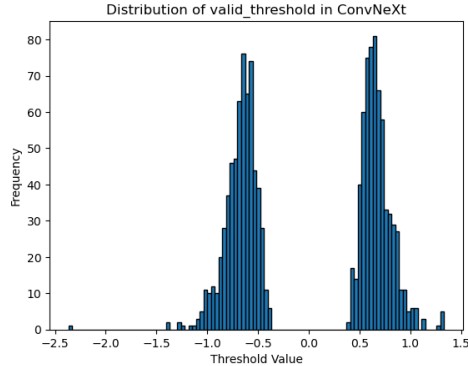

Figure 6: Distribution of learned thresholds for valid predicates in ConvNeXt.

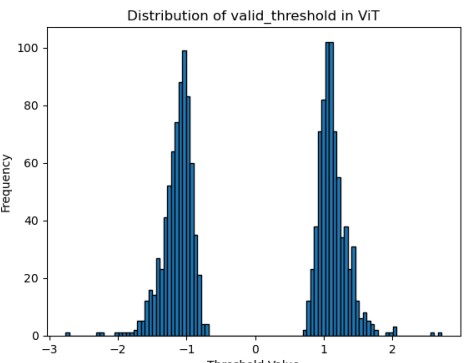

Figure 7: Distribution of learned thresholds for valid predicates in ViT.

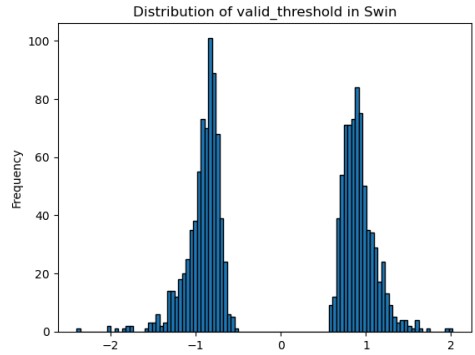

Figure 8: Distribution of learned thresholds for valid predicates in Swin.

# D  ADDITIONAL DETAILS ON ASSESSING RULE-BASED EXPLANATIONS

## D.1  MERTICS

For completeness, we report the metrics used to evaluate global logical rule–based explanations:

- **Number of valid predicates:** Count of predicates that take both `True` and `False` values on the evaluation set under the learned thresholds.
- **Explanation complexity:** Average number of predicates in the selected explanation per image (equivalently, the average number of literals in the satisfied DNF clause).
- **Coverage:** Fraction of images for which VISIONLOGIC returns a valid explanation (e.g., at least one rule is satisfied or a score-based explanation is produced).
- **Fidelity (covered):** Among covered images, the percentage for which VISIONLOGIC's predicted class matches the base model's predicted class.
- **Top-1 accuracy (covered):** Among covered images with ground-truth labels, the percentage for which VISIONLOGIC's top-1 class equals the ground-truth class.
- **Top-5 accuracy (covered):** Among covered images with ground-truth labels, the percentage for which VISIONLOGIC's top-5 class equals the ground-truth class.

## D.2  PROBING ROBUSTNESS OF VISION MODELS

Although VISIONLOGIC is trained exclusively on positive examples, it still correctly identifies a non-trivial fraction of images misclassified by the neural networks. This is evident from the gap between *Fidelity (covered)* and *Top-1 accuracy (covered)*: VISIONLOGIC can match erroneous model predictions with rules, offering insights into misclassification causes and helping probe robustness under perturbations and adversarial settings.

We investigate the success of adversarial attacks through the lens of local explanations generated by VISIONLOGIC. From a logical rule perspective, misclassification typically occurs when (a) the top-ranked predicates of the ground truth class are deactivated, and (b) predicates associated with other classes become active. While these often co-occur, we define (a) as the root cause—denoted as a *Type A* cause—if it alone can alter the prediction without (b). Similarly, we define a *Type B* cause when (b) alone is sufficient to induce misclassification. These two causes are attack-agnostic, enabling us to understand the underlying logic behind different attacks.

We provide concrete examples in Figure 9. The original image follows the rule $p_{669} \wedge p_{844} \wedge p_{489} \Rightarrow$ *"jay"*. After the PGD (Madry et al., 2018) attack, the rule becomes $p_{1220} \wedge p_{489} \wedge p_{537} \wedge p_{844} \Rightarrow$ *"tray"*, which is a Type B cause, as introducing the new predicate $p_{1220}$ significantly increases the explanation score for the class *"tray"*. The Gaussian Noise attack yields $p_{2032} \wedge p_{1074} \wedge p_{2028} \cdots \wedge p_{669} \wedge p_{844} \wedge p_{489} \Rightarrow$ *"badger"*, introducing many new predicates, which is a Type B cause. The Pixelation attack results in $p_{2028} \wedge p_{376} \wedge p_{1940} \cdots \wedge p_{211} \Rightarrow$ *"black grouse"*, deactivating all original predicates, thus exhibiting a Type A cause.

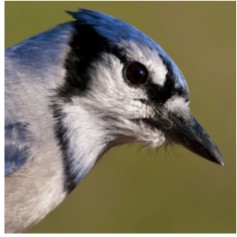 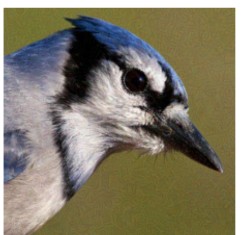 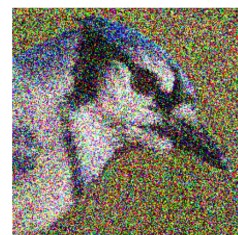 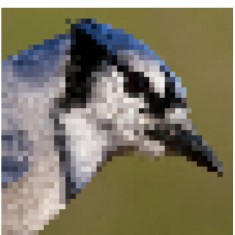

Figure 9: This displays the original image, followed by the images after PGD, Gaussian noise, and Pixelate attacks from left to right.

To conduct a systematic evaluation, we randomly sample 1,000 images from the ImageNet validation set, applying a successful attack to each image once using Projected Gradient Descent (PGD) (Madry

et al., 2018), Gaussian noise (Goodfellow et al., 2016), and Pixelate attacks (Engstrom et al., 2019), respectively.

Table 3: Statistics of root causes under various adversarial attacks.

| Attack | ResNet | | ConvNet | | ViT | | Swin | |
|--------|--------|--------|---------|--------|--------|--------|--------|--------|
| | Type A | Type B | Type A | Type B | Type A | Type B | Type A | Type B |
| Gaussian | 79 | 921 | 116 | 884 | 149 | 851 | 109 | 891 |
| Pixelate | 214 | 786 | 329 | 671 | 267 | 733 | 298 | 702 |
| PGD | 377 | 623 | 561 | 439 | 653 | 347 | 612 | 388 |

Table 3 presents the statistics of root causes across different adversarial attacks. The Gaussian noise attack tends to affect all active predicates simultaneously, often activating new predicates, which is primarily explained by Type B causes. The Pixelate attack operates similarly but has a higher chance of deactivating existing predicates, leading to more Type A causes, as it dilutes the fine-grained details of regions attended by the predicates. Finally, the PGD attack is the most efficient and selects either Type A or Type B causes to achieve the quickest result. For ResNet, which generates shorter explanations, it favors introducing new predicates. For transformers, which use more predicates in their explanations, it leans more toward deactivations. We aim to investigate more attacks and defenses in future work.

## E   ADDITIONAL OBSERVATIONS ON PREDICATES

$p_{908}$  $p_{(498,\,-)}$  $p_{219}$  $p_{(312,\,+)}$

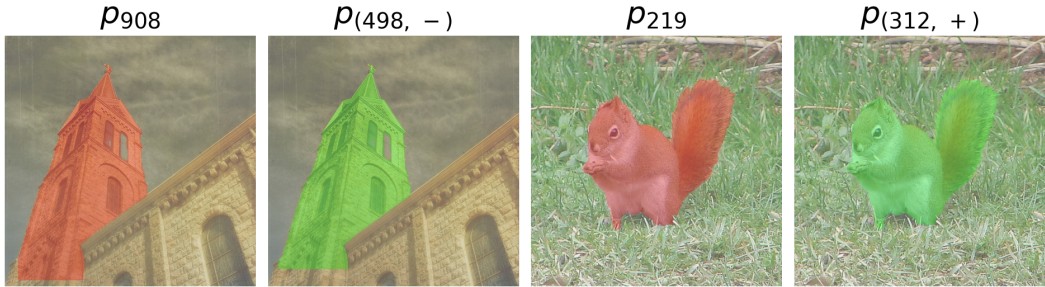

Figure 10: Top-ranked predicates often capture global structure. Red: ResNet; green: ViT.

**Top-ranked predicates encode global structure.**   Beyond the local and modular concepts discussed above, some predicates capture *global* object structure. Figure 10 illustrates two classes (*church*, *squirrel*), where $p_{908}$, $p_{(498,-)}$, $p_{219}$, and $p_{(312,+)}$ (ResNet/ViT) encode the entire church or squirrel. Causally, masking any single part does not deactivate these predicates, but masking the whole object does. These global predicates are typically top-ranked within their classes and tend to be more class-specific, whereas local predicates are more frequently shared across classes. This pattern holds for both CNNs and Transformers and suggests a potential avenue for further rule simplification.

**Predicates robustly identify visual concepts across variations in appearance.**   The human visual system can recognize objects belonging to the same concept despite significant differences in appearance, such as color and shape. Interestingly, our learned predicates exhibit a similar capability. For example, in the third row of Figure 4, the concept *"bird's head"* appears from both the front and the side, yet $p_{1890}$ consistently attends to it. Similarly, $p_{1878}$ captures varying forms, sizes, and angles of *"church tops"*.

**Sensitivity to location changes.**   Some predicates appear to be location-invariant, while others are not, as illustrated in Figure 11. Predicate $p_{170}$ in the top row remains active even when the *"fox's ears"* are shifted across the third, fourth, and fifth images. In contrast, predicate $p_{1916}$ in the bottom row, which encodes *"squirrel's tail"*, is deactivated once the tail moves away from its original position.

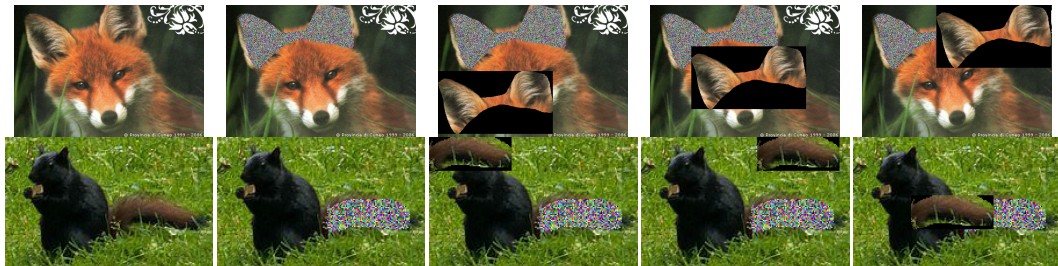

Figure 11: Sensitivity to spatial changes. The predicate encoding "fox's ears" appears to be location-invariant, whereas the predicate encoding "squirrel's tail" is sensitive to positional shifts.

## F  HUMAN EVALUATION

### F.1  EXPERIMENT SETUP

We describe the evaluation setup in detail as follows.

**Participants.**  We recruited participants from Prolific (Prolific, 2024), a popular online platform for human evaluation of research projects. The users recruited have high qualifications as we requested a task acceptance rate greater than 98%. Our questionnaire is approved by the Social Sciences, Humanities & Education REB at University X, and all users provided informed consent before they started the experiment. The questionnaire is designed to be completed in 8-10 minutes and users who passed the screening received USD$ 2.00 upon completion.

**Statistics.**  For the Husky vs. Wolf scenario, $n = 161$ participants passed screening and filtering, respectively $n = 31, 25, 40, 35, 30$ for control, baseline, ACE, CRAFT and VISIONLOGIC.

For the Otter vs. Beaver scenario, $n = 114$ participants passed screening and filtering, respectively $n = 22, 22, 27, 25, 18$ for control, baseline, ACE, CRAFT, and VISIONLOGIC.

For the Kit Fox vs. Red Fox scenario, $n = 190$ participants passed screening and filtering, respectively $n = 35, 35, 45, 40, 35$ for control, baseline, ACE, CRAFT, and VISIONLOGIC.

**Study design.**  We followed the experimental design proposed by Colin et al. (2022), which quantitatively assesses to what degree a concept-based explanation method can help a human observer understand the behavior of an AI model. Each participant is only tested on a single condition (control, baseline, ACE, CRAFT, or VISIONLOGIC) to avoid possible experimental confounds. *For fairness, we compare only the causally grounded concepts produced by VISIONLOGIC, excluding any rule-structure information, against the four other conditions.*

The screening, training, and testing phases are exactly as described in Colin et al. (2022). After granting consent (shown in Figure 12), participants are presented with detailed instructions and the study goal: learning to predict which class *an AI model will predict* for a given image. They start with a practice session that shows simple images along with their explanations and model prediction, followed by some unseen images without explanations, where they are expected to answer the model's prediction correctly. Participants who failed this session were not allowed to proceed with the study. They are then tested on their understanding of the study goal by a short quiz (exactly formulated as in Colin et al. (2022)). Again, participants who failed this session were not allowed to proceed.

After the above two screening phases, participants went through 3 sessions of training phases (5, 10, and 15 images, respectively, with explanations and model decisions) followed by a testing phase (7 new images without explanation). The answers for the testing phases are collected to compute the main result. Figure 13 shows an example of the training phase. As described in Colin et al. (2022), we implemented a reservoir (Figure 14) for training images as a reference point for participants during the testing phase. The last test image of each session is an image from the reservoir, and participants who incorrectly answered the last test question were filtered out.

Learn to predict AI model decisions by studying explanations

**AI Explanation Study Instructions**

**Your Goal:** Learn to predict what an AI model will output by studying its explanations.

**Important Guidelines:**

- **Focus on the AI's behavior** - not what you think is "correct"
- **Study the explanations carefully** - they show what the AI is looking at
- **During training:** You'll see images + explanations + AI predictions
- **During testing:** You'll see only images and must predict what the AI will say
- **Use the reference panel** - you can look back at training examples during tests
- **If you do not see an explanation image** - you are in the control group, please try your best to predict AI's decision

**Payment:** You must pass all screening questions and attention checks to receive payment.

**Participant Consent Form**

### Explainable AI: human interpretation of machine learning systems

**Invitation to Participate**

You are invited to participate in a short online research study conducted by researchers at the ▓▓▓▓▓▓▓▓▓▓▓▓. The purpose of this study is to better understand how people interpret the explanations provided by machine learning models. Your participation is entirely voluntary.

**What Will You Be Asked to Do?**

If you agree to participate:

- You will complete a short series of visual tasks.
- The study will take approximately 5–7 minutes.

**Compensation**

You will be compensated ▓▓▓▓▓▓▓▓▓▓▓▓▓▓▓▓▓▓▓▓▓▓▓▓▓▓▓▓, based on the estimated time commitment and in accordance with fair-pay guidelines (approx. $15/hour equivalent). Partial payment may be issued if you begin but do not complete the full task.

**Risks and Benefits**

There are no known risks beyond those of normal online activity. While there are no direct personal benefits, your participation will help improve understanding of how people interact with machine learning explanations, which may benefit future AI system design.

**Voluntary Participation and Withdrawal**

Participation is voluntary. You may withdraw at any time by closing your browser. If you do not complete the task, your data will not be saved. If you complete and submit the task, your data will be anonymized and cannot be withdrawn, as it will no longer be linked to you.

**Confidentiality**

- No personally identifying information will be collected.
- All data will be stored securely and analyzed in aggregate.
- Results may be published or presented, but individual participants will not be identifiable.

**Questions or Concerns**

If you have any questions about the study, you may contact the student researcher at: ▓▓▓▓▓▓▓▓▓▓▓▓

If you have questions about your rights as a research participant, please contact the ▓▓▓▓▓▓▓▓▓▓▓▓ ▓▓▓▓▓▓▓▓▓▓▓▓

**Consent Statement**

By checking the box below and proceeding to the task, you confirm:

- You are 18 years of age or older;
- You have read and understood the information above;
- You voluntarily consent to participate in this study.

○ **I consent to participate in this study**

○ **I do not consent to participate in this study**

Figure 12: The online questionnaire begins with a consent form.

## F.2 STATISTICAL TEST RESULTS FOR SIGNIFICANCE

We provide statistical test results in Figure 15, 16, 17, 18.

**Training Example 1 of 15**

**AI Prediction: Class Wolf**

Continue

Figure 13: The online questionnaire displaying an example of the training session with the original image, the explanation, and the model prediction.

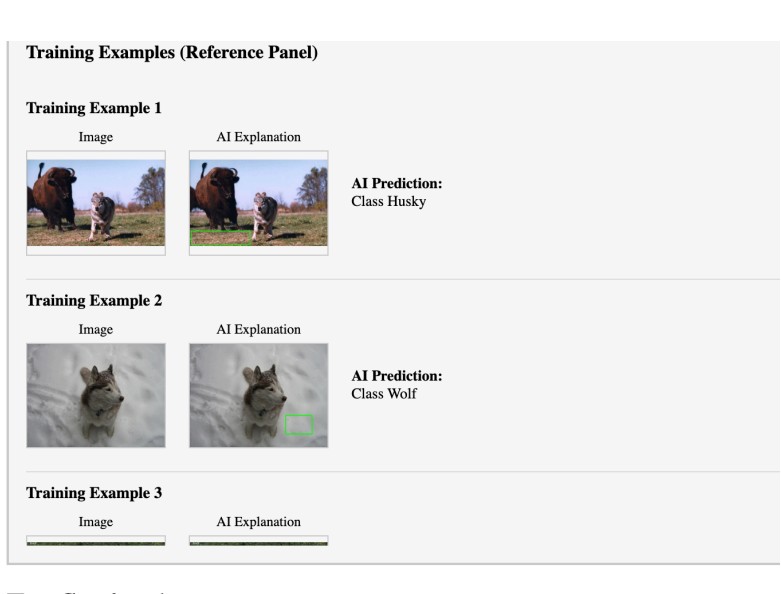

## Test Session 1

Now predict what the AI will say for these images. You can refer to the training examples above.

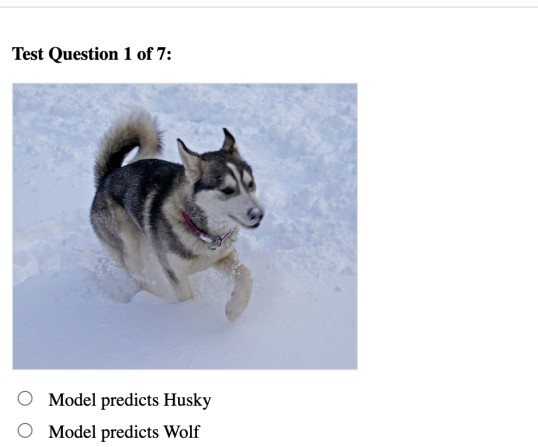

Figure 14: The online questionnaire displaying the testing session with a reservoir containing all examples during the training phase.

```
Starting analysis for EXP1
================================================================
Loading exp1 data files...
========================================================
✓ Control : 31 participants loaded from Control_exp1_individual_results.csv
✓ Baseline: 25 participants loaded from Baseline_exp1_individual_results.csv
✓ ACE     : 40 participants loaded from ACE_exp1_individual_results.csv
✓ CRAFT   : 35 participants loaded from CRAFT_exp1_individual_results.csv
✓ LOGIC   : 30 participants loaded from LOGIC_exp1_individual_results.csv

Total participants loaded: 161
Methods found: ['ACE', 'Baseline', 'CRAFT', 'Control', 'LOGIC']
========================================================
NORMALITY TEST (Shapiro-Wilk)
========================================================
H0: Data is normally distributed
H1: Data is not normally distributed
Significance level: 0.05
-------------------------------------------------------------
Control | n=31 | W=0.9372 | p=0.0689 | Normal
Baseline | n=25 | W=0.9521 | p=0.2790 | Normal
ACE     | n=40 | W=0.9404 | p=0.0357 | Not Normal
CRAFT   | n=35 | W=0.9376 | p=0.0473 | Not Normal
LOGIC   | n=30 | W=0.8189 | p=0.0001 | Not Normal

Overall normality assumption:  ✗ VIOLATED
========================================================
HOMOGENEITY OF VARIANCE TEST (Levene's Test)
========================================================
H0: All groups have equal variances
H1: At least one group has different variance
Significance level: 0.05
-------------------------------------------------------------
Levene's statistic: 1.3810
p-value: 0.2430
Homogeneity assumption: ✓ MET

Group variances:
Method   | n | Variance |  SD  |  SE
-----------------------------------------------
Control | 31 |  0.0334 | 0.1827 | 0.0328
Baseline | 25 |  0.0385 | 0.1962 | 0.0392
ACE     | 40 |  0.0593 | 0.2435 | 0.0385
CRAFT   | 35 |  0.0317 | 0.1780 | 0.0301
```

Figure 15: Complete statistical test results for scenario 1, page 1.

```
LOGIC    | 30 |   0.0299 |   0.1730 |   0.0316

==========================================================
NON-PARAMETRIC ANALYSIS
==========================================================
==========================================================
KRUSKAL-WALLIS TEST (Non-parametric)
==========================================================
H0: All group distributions are identical
H1: At least one group distribution differs
Significance level: 0.05
-------------------------------------------------------------
H-statistic: 25.8440
p-value: 0.000034
Degrees of freedom: 4
Result: ✓ SIGNIFICANT

→ Significant differences found between method distributions.
→ Proceeding to Dunn's post-hoc test...

==========================================================
DUNN'S TEST (Non-parametric post-hoc)
==========================================================
Pairwise p-values (Bonferroni corrected):
         ACE Baseline  CRAFT  Control  LOGIC
ACE      1.0000   1.0000  0.5956  1.0000  0.0241
Baseline 1.0000   1.0000  1.0000  1.0000  0.0303
CRAFT    0.5956   1.0000  1.0000  1.0000  0.0000
Control  1.0000   1.0000  1.0000  1.0000  0.0004
LOGIC    0.0241   0.0303  0.0000  0.0004  1.0000

Pairwise Comparisons Summary (Bonferroni corrected, α = 0.05):
================================================================
ACE      vs Baseline | p:  1.0000
ACE      vs CRAFT    | p:  0.5956
ACE      vs Control  | p:  1.0000
ACE      vs LOGIC    | p:  0.0241 ***
Baseline vs CRAFT    | p:  1.0000
Baseline vs Control  | p:  1.0000
Baseline vs LOGIC    | p:  0.0303 ***
CRAFT    vs Control  | p:  1.0000
CRAFT    vs LOGIC    | p:  0.0000 ***
Control  vs LOGIC    | p:  0.0004 ***
```

Figure 16: Complete statistical test results for scenario 1, page 1.

```
Starting analysis for EXP2
=================================================================
Loading exp2 data files...
=================================================================
✓ Control : 22 participants loaded from Control_exp2_individual_results.csv
✓ Baseline: 22 participants loaded from Baseline_exp2_individual_results.csv
✓ ACE     : 27 participants loaded from ACE_exp2_individual_results.csv
✓ CRAFT   : 25 participants loaded from CRAFT_exp2_individual_results.csv
✓ LOGIC   : 18 participants loaded from LOGIC_exp2_individual_results.csv

Total participants loaded: 114
Methods found: ['ACE', 'Baseline', 'CRAFT', 'Control', 'LOGIC']
=================================================================
NORMALITY TEST (Shapiro-Wilk)
=================================================================
H0: Data is normally distributed
H1: Data is not normally distributed
Significance level: 0.05
-------------------------------------------------------------
Control  | n=22 | W=0.7308 | p=0.0000 | Not Normal
Baseline | n=22 | W=0.6701 | p=0.0000 | Not Normal
ACE      | n=27 | W=0.8543 | p=0.0014 | Not Normal
CRAFT    | n=25 | W=0.8510 | p=0.0018 | Not Normal
LOGIC    | n=18 | W=0.6087 | p=0.0000 | Not Normal

Overall normality assumption: ✗ VIOLATED
=================================================================
HOMOGENEITY OF VARIANCE TEST (Levene's Test)
=================================================================
H0: All groups have equal variances
H1: At least one group has different variance
Significance level: 0.05
-------------------------------------------------------------
Levene's statistic: 2.5161
p-value: 0.0455
Homogeneity assumption: ✗ VIOLATED

Group variances:
Method   | n | Variance |  SD   |  SE
----------------------------------------------
Control  | 22 |  0.0230 | 0.1517 | 0.0323
Baseline | 22 |  0.0282 | 0.1678 | 0.0358
ACE      | 27 |  0.0222 | 0.1491 | 0.0287
CRAFT    | 25 |  0.0123 | 0.1111 | 0.0222
```

Figure 17: Complete statistical test results for scenario 2, page 1.

```
LOGIC    | 18 |   0.0011 |  0.0332 |   0.0078

=========================================================
NON-PARAMETRIC ANALYSIS
=========================================================

=========================================================
KRUSKAL-WALLIS TEST (Non-parametric)
=========================================================
H0: All group distributions are identical
H1: At least one group distribution differs
Significance level: 0.05
-------------------------------------------------------------
H-statistic: 13.5628
p-value: 0.008830
Degrees of freedom: 4
Result: ✓ SIGNIFICANT

→ Significant differences found between method distributions.
→ Proceeding to Dunn's post-hoc test...
=========================================================
DUNN'S TEST (Non-parametric post-hoc)
=========================================================
Pairwise p-values (Bonferroni corrected):
          ACE  Baseline  CRAFT  Control  LOGIC
ACE     1.0000   1.0000 1.0000   1.0000 0.0045
Baseline 1.0000  1.0000 1.0000   1.0000 0.1556
CRAFT   1.0000   1.0000 1.0000   1.0000 0.0309
Control 1.0000   1.0000 1.0000   1.0000 0.1463
LOGIC   0.0045   0.1556 0.0309   0.1463 1.0000

Pairwise Comparisons Summary (Bonferroni corrected, α = 0.05):
===================================================================
ACE      vs Baseline | p:  1.0000
ACE      vs CRAFT    | p:  1.0000
ACE      vs Control  | p:  1.0000
ACE      vs LOGIC    | p:  0.0045 ***
Baseline vs CRAFT    | p:  1.0000
Baseline vs Control  | p:  1.0000
Baseline vs LOGIC    | p:  0.1556
CRAFT    vs Control  | p:  1.0000
CRAFT    vs LOGIC    | p:  0.0309 ***
Control  vs LOGIC    | p:  0.1463
```

Figure 18: Complete statistical test results for scenario 2, page 2.

