# OpenReview forum: "VisionLogic: From Neuron Activations to Causally Grounded Concept Rules for Vision Models"
_ICLR.cc/2026/Conference — ICLR 2026 Conference Withdrawn Submission_

### Official Review · Reviewer_tFjU · 2025-10-30

**Soundness:** 2
**Presentation:** 3
**Contribution:** 3
**Rating:** 6
**Confidence:** 3

**Summary:**

This paper focuses on the problem of concept-based explanations for vision models. While existing concept-based methods offer semantic interpretability, they often rely on correlations rather than causal reasoning, which can lead to spurious or misleading explanations. To address these limitations, the paper introduces a neural–symbolic framework that converts neuron activations into interpretable predicates, extracts global logical rules over these predicates, and grounds them to visual concepts. This approach enables the discovery of causally validated and human-understandable concepts. Experiments on CNNs and ViTs demonstrate that the proposed method retains high predictive performance while producing compact and interpretable rules. Large-scale human evaluations are also provided to show the enhanced understanding of model behavior compared to prior concept-based baselines.

**Strengths:**

1. The paper is well-written and provides clear visualizations.
2. The paper proposes a new perspective on concept-based explanations from a neural-symbolic reasoning approach, which can offer causally validated concepts for more trustworthy explanations.
3. The paper conducts extensive human evaluation to validate the effectiveness of the generated concept explanations in enhancing human understanding and decision making.

**Weaknesses:**

1. The paper appears to miss some recent baselines in concept-based explanations, such as [1, 2, 3], which makes the performance improvement achieved through neural symbolic reasoning less convincing.
2. The methodology involves multiple stages, different components, and various hyperparameters; however, the paper lacks sufficient ablation studies to analyze the impact of each factor.
3. The paper evaluates performance only on a single image dataset from general domains. However, it would be beneficial to include some domain-specific datasets, such as medical imaging datasets, to provide a more comprehensive evaluation and demonstrate the practicality of the proposed concept explanations for decision-making.

[1] Explain Any Concept: Segment Anything Meets Concept-Based Explanation, NeurIPS 2023.

[2] A Holistic Approach to Unifying Automatic Concept Extraction and Concept Importance Estimation, NeurIPS 2023.

[3] Explainable Concept Generation through Vision-Language Preference Learning for Understanding Neural Networks’ Internal Representations, ICML 2025.

**Questions:**

1. How does the performance of the proposed neural–symbolic explanation method compare with other baselines?
2. How do the different components affect performance, such as the segmentation-based refinement step using Mask R-CNN and SAM?

---

### Official Review · Reviewer_ZJEN · 2025-10-30

**Soundness:** 2
**Presentation:** 3
**Contribution:** 2
**Rating:** 4
**Confidence:** 3

**Summary:**

The paper introduces VISIONLOGIC, a neuro-symbolic interpretability framework for vision models. It learns per-channel thresholds to convert last-layer activations into a binary predicate vocabulary, induces global, class-wise logical rules over these predicates with a ranking-based inference score, and causally grounds each predicate into a human-readable visual concept via ablation-based localization with iterative box refinement and segmentation masks.

**Strengths:**

* Predicates are localized by occlusion-driven search and sufficiency checks, then refined via segmentation; the pipeline operationalizes necessity/sufficiency rather than relying on correlation only.
* The paper observes shorter rules for CNNs and longer, compositional clauses for Transformers, aligning with recent findings and offering a hypothesis for predicate usage differences.

**Weaknesses:**

* The necessity/sufficiency status of a region depends on the occluder (noise/blur/mean/black/white) and crop-paste context. The paper notes other replacements can work and even that blurring often performs best, but it does not quantify agreement across intervention types.
* Rule stability. Thresholds and learned rules are post-hoc; the paper does not report seed-to-seed or fine-tuning stability.
* The final mask uses off-the-shelf segmenters and then re-validates causality. This is sensible, but the choice (SAM, ISNet, Mask R-CNN) may bias boundaries and occasionally produce over fragmented parts.
* How do coverage and fidelity change on ImageNet-R or style-transferred images (backgrounds, textures)?

**Questions:**

Please refer to the Weaknesses.

---

### Official Review · Reviewer_sGhN · 2025-10-31

**Soundness:** 2
**Presentation:** 4
**Contribution:** 3
**Rating:** 4
**Confidence:** 3

**Summary:**

The authors propose a method that bridges concept-based explanations for computer vision tasks with predicate-/logic-based ones. The approach is architecture-agnostic and post-hoc. Training follows a two-stage pipeline:

Train a surrogate classifier on the base model predictions to estimate predicate thresholds and temperature parameters.

Train a predicate-to-visual-concept linker that maps visual patches (segmentation-based) to the learned predicates.
Experiments include a large-scale user study showing the efficacy of the method.

**Strengths:**

The paper is an interesting fusion of existing ideas. The combination of concept-based learning (supported with existing semantic segmentation methods) and logical frameworks is original and useful. The presentation is excellent - concise, well-organized, and easy to follow.

Another major strength is the extensive user study, which is designed around a solid XAI evaluation framework [1]. Overall, the method and results are convincing and well-presented, meeting ICLR standards.

[1] Colin, Julien, et al. "What I cannot predict, I do not understand: A human-centered evaluation framework for explainability methods." NeurIPS 35 (2022): 2832–2845.

**Weaknesses:**

While I genuinely appreciate this work and believe it would make a good ICLR contribution, it has one major shortcomming: the causal framing is simply not substantiated.

The paper repeatedly claims to be "causally grounded" or "causally validated." The title, abstract, and introduction all position this as a paper in causal inference. But there's no actual causal framework here: no structural causal model, no assumptions about confounding, no causal graph, no identifiability reasoning, nothing that grounds it in the causal inference literature.

What the authors call "causal validation" is a sensitivity analysis: turning regions on and off, observing changes in predictions. It's useful, informative, and valid as a faithfulness test; and it is NOT causal. Calling it causal sets a dangerous precedent, because then any perturbation or ablation analysis could be sold as "causal."
If the authors reframed their method honestly, e.g., "perturbation-grounded concept validation", "necessary-region validation", or simply "ablation-based faithfulness tests", it would stand on its own merits. It's already a strong contribution. It doesn't need causal marketing.

Another issue is that the evaluation focuses almost entirely on the visual side of explanations. The user study measures how interpretable the visual outputs are, but doesn't say much about the logical predicates or rule-based explanations themselves. Since the method is supposed to connect logic and vision, it's surprising that no human or quantitative evaluation targets the logical rules directly. Maybe the rules do not need to be human-interpretable, but the merit of these rules remains questionable then.

Also, the baselines could be more diverse. The paper includes concept-based and segmentation-based comparisons (and the saliency/control method), but lacks the standard pixel-level interpretability methods (GradCAM, Occlusion, etc.) that would situate results against the broader XAI landscape, especially given that [1] showed these methods often perform competitively in user studies.


A more detailed ablation study would also help. For example:

- How much do predicate thresholds and temperature affect concept stability?

- How does varying the segmentation method influence the link between predicates and visual regions?

- What happens if the predicates are fixed but visual linking changes (in the second training stage)?


Such analyses would clarify how robust each stage of the proposed pipeline really is.

To be clear: This is a good manuscript. It's technically sound, well-executed, and could be impactful if framed correctly. But the current causal language overreaches. As it stands, the paper is a strong contribution to faithfulness and interpretability, not to causal inference.

The reason I cannot vote for accept is that the framing is misleading. If the authors remove the causal rhetoric and stick to what they've actually shown, I will update my score accordingly.

For reference, the kind of "causal" deep learning work this paper gestures toward includes explicit causal assumptions, identifiable mechanisms, counterfactuals, or domain generalization arguments - see [2-5]. This paper doesn't engage with that level of rigor, which is fine. Just don't call it causal.

[1] Colin, Julien, et al. "What I cannot predict, I do not understand: A human-centered evaluation framework for explainability methods." NeurIPS 35 (2022): 2832–2845.
[2] Schulte, Rickmer, David Rügamer, and Thomas Nagler. "Adjustment for Confounding using Pre-Trained Representations." ICML 2025.
[3] Madras, David, et al. "Fairness through Causal Awareness." FAT* 2019.
[4] Pettersson, Markus B., et al. "Time Series of Satellite Imagery Improve Deep Learning Estimates of Neighborhood-Level Poverty in Africa." IJCAI 2023.
[5] Louizos, Christos, et al. "Causal Effect Inference with Deep Latent-Variable Models." NeurIPS 2017.

**Questions:**

What motivated the causal framing? How do the authors see their work to be different from standard correlational approaches?

Why not include more classical pixel-level baselines (GradCAM, Saliency, Occlusion) for completeness?

Can the authors provide their implementation? This would make the contribution more impactful for the community.

---

### Official Review · Reviewer_yjw5 · 2025-11-01

**Soundness:** 2
**Presentation:** 4
**Contribution:** 3
**Rating:** 4
**Confidence:** 4

**Summary:**

The paper provides a causally validated explainability framework termed VisionLogic for visual interpretability. Specifically, VisionLogic has 3 stages: (i) activations are used to construct predicates, (ii) predicates construct inference rules and scores, and (iii) use occlusion ablation to causally validate. The authors address critical gap in concept-based interpretability through causal validation. Additionally, the paper is the first work providing explicit global logical rules at the full ImageNet scale for 4 architectures including transformers.

**Strengths:**

1. **Clarity**: The paper has a clear structure that introduces concepts to the readers in a readable manner. The notations are dense in some places, but not overtly so. The usage of figures and appendices is mostly excellent (except D.1 clarity which I question below). The paper is overall well-written.

2. **Rigorous Human Evaluation**: 465 participants on 3 scenarios with proper statistical testing.

3. **Sound Technical Framework**: The 3 stages are together well-designed. The methods generalizes across CNNs and Transformers.

4. **Results Scale and Performance**: As the authors state, VisionLogic is the first explainability framework to train and validate on the full ImageNet-1k dataset.

5. **Comprehensive Evaluation**: The authors go above and beyond normal evaluation setups by including adversarial robustness analysis (D.2), multiple segmentation backends and detailed ablations. This is highly appreciated.

**Weaknesses:**

**Technical soundness**:

1. The following sentence is, in my opinion, concerning (Line 153): "For clarity, all classwise statistics in Section 3 use only training examples correctly classified by the base model, preventing predicate learning from contamination by misclassified instances." In datasets like ImageNet, it can be argued that the networks have learned everything there is to learn. Hence, the proposed predicates are entirely dependent on a good model. How good will the VisionLogic Framework be in datasets and networks where this is not the case? In other words, how robust are the activation predicates? I believe this is an important question to address since it's the first step in the entire framework.

Please note: The robustness I am referring to is different from Section D.2. My concern is with learning the predicates themselves, rather than utilizing *well-learned* predicates for robust explanations.

2. Additionally, utilizing predicates from a learned network based on correct predicates may lead to confirmation bias. The networks may have learned incorrect (correlation) rules which will just be confirmed later. Recent work [1] has shown that correct/incorrect results do not always provide same trends, specifically when evaluation uses occlusion.

[1] Prabhushankar et al. "Voice: Variance of induced contrastive explanations to quantify uncertainty in neural network interpretability." IEEE Journal of Selected Topics in Signal Processing 19.1 (2024): 19-31.

**Result Discussion**

1. I am unsure how to read the metrics in Table 2. App D.1 does not easily clear it up. For instance, in Table 2, ResNet Top-1 accuracy is 69%. Based on D.2, have the authors: (i) chosen N samples among the 50,000 validation images, (ii) among the N samples, M are covered (at least 1 concept activated), and (iii) 69% of M samples' prediction from the original classification layer match the predicate inference? I have similar questions about all other metrics as well.

2. I feel the complexity is a major issue (even though the authors acknowledge this in Line 438). What does > 30 complexity mean? Qualitatively showcasing this is important. Is all the image covered? This goes back to my earlier question about the predicates themselves. Maybe other fine-grained classification datasets may tell the authors more.

3. The paper can use more negative qualitative results. For instance, showcasing the images which are not covered (10-20% in Table 2) can be very informative.

4. Other qualitative results include showcasing polysemanticilty. It will be interesting to see and analyze common features and common features between networks

**Minor weaknesses**:

1. In Eq.1, R^c needs to be defined as ordered rank. Currently the first definition seems to be in Line 240

2. Line 362: identica -> identical

**Questions:**

Please see the weaknesses. I would be willing to increase my score if the following questions are answered:

1. How robust are the predicates?

2. How would the predicates and results change when including incorrect results (maybe hard to find on ImageNet training set)?

3. Could you please clarify the metrics in Table 2

4. Is there a focused human study where 50+ predicate explanations are shown to be interpretable?

5. Are the uncovered images (i) concentrated in specific classes, or (ii) have lower prediction confidence, or (iii) represent out-of-distribution cases, or (iv) incorrect ground truths, or (v) just something fundamentally impossible to characterize in ImageNet?

6. What percentage of predicates encode multiple distinct concepts, and how does this affect rule trustworthiness and interpretability?

---

### Author Response · Authors · 2025-11-13
**Appreciation for Reviewer Effort**

We sincerely thank all reviewers for their time, effort, and detailed feedback. We truly appreciate the thoughtful and constructive comments provided. They have been extremely valuable for improving the clarity, technical rigor, and overall presentation of our work.

We have decided to withdraw the submission and will carefully refine the paper to further improve its quality. All reviewer feedback will be taken seriously and incorporated into the next revision.

Thank you again for your contributions to strengthening our research.

---

### Note · Authors · 2025-11-13

I have read and agree with the venue's withdrawal policy on behalf of myself and my co-authors.